# Siwi cooperates with Par-1 kinase to resolve the autoinhibitory effect of Papi for Siwi-piRISC biogenesis

Hiromi Yamada[1], Kazumichi M. Nishida[1], Yuka W. Iwasaki [2,3], Yosuke Isota[1], Lumi Negishi[4] & Mikiko C. Siomi [1✉]

*Bombyx* Papi acts as a scaffold for Siwi-piRISC biogenesis on the mitochondrial surface. Papi binds first to Siwi via the Tudor domain and subsequently to piRNA precursors loaded onto Siwi via the K-homology (KH) domains. This second action depends on phosphorylation of Papi. However, the underlying mechanism remains unknown. Here, we show that Siwi targets Par-1 kinase to Papi to phosphorylate Ser547 in the auxiliary domain. This modification enhances the ability of Papi to bind Siwi-bound piRNA precursors via the KH domains. The Papi S547A mutant bound to Siwi, but evaded phosphorylation by Par-1, abrogating Siwi-piRISC biogenesis. A Papi mutant that lacked the Tudor and auxiliary domains escaped coordinated regulation by Siwi and Par-1 and bound RNAs autonomously. Another Papi mutant that lacked the auxiliary domain bound Siwi but did not bind piRNA precursors. A sophisticated mechanism by which Siwi cooperates with Par-1 kinase to promote Siwi-piRISC biogenesis was uncovered.

[1] Department of Biological Sciences, Graduate School of Science, The University of Tokyo, Tokyo 113-0032, Japan. [2] Department of Molecular Biology, Keio University School of Medicine, Tokyo 162-8582, Japan. [3] Japan Science and Technology Agency, Precursory Research for Embryonic Science and Technology, Saitama 332-0012, Japan. [4] Central Laboratory, Institute for Quantitative Biosciences, The University of Tokyo, Tokyo 113-0032, Japan. ✉email: siomim@bs.s.u-tokyo.ac.jp

P IWI-interacting RNAs (piRNAs) repress transposons to protect germline genomes from DNA damage caused by random movement of transposons within a genome[1–3]. To fulfill this role, piRNAs stoichiometrically assemble the piRNA-induced silencing complex (piRISC) with PIWI proteins. Dysfunction of piRISC causes transposon derepression, leading to failures in gonadal development and infertility[4].

The *Bombyx* genome contains two *PIWI* genes, *Siwi* and *Ago3*[5], and BmN4 cells, which are cultured germ cells derived from *Bombyx* ovaries, express both PIWI proteins[5,6]. In BmN4 cells, Siwi and Ago3 are loaded with piRNAs, localized to the cytoplasm, and exert endonuclease activity, called slicer, to cleave target RNAs complementary to the bound piRNAs, resulting in post-transcriptional repression of the target genes[5,6]. Many other organisms such as *Drosophila* and mouse have nuclear-localized piRISCs in addition to cytoplasmic piRISCs, which induce heterochromatin at the target gene loci to block transcription[7]. *Bombyx* lacks nuclear-localized piRISCs, thus all piRISC-mediated transposon repression is PIWI slicer-dependent[6,8].

In BmN4 cells, Siwi-loaded piRNAs are antisense biased against transposon mRNAs, and therefore Siwi-piRISC is able to cleave transposon mRNAs[6]. Cleaved RNAs are subsequently used to generate Ago3-piRISC. Thus, Ago3-loaded piRNAs are sense biased and Ago3-piRISC cleaves antisense transcripts of transposons. These cleaved RNAs are then used to produce Siwi-piRISC[6]. This slicer-dependent, mutual RNA cleavage operated by the two PIWI proteins is known as the ping-pong cycle, which exhausts transposon RNA transcripts intracellularly, resulting in piRNA amplification and effective transposon repression[9–11].

The Ago3 slicer-dependent Siwi-piRISC production occurs in nuage termed Ago3 bodies[6]. Nuage is a germ cell-specific, perinuclear non-membranous organelle that is conserved in a wide range of organisms[12,13]. However, in BmN4 cells, Ago3-independent Siwi-piRISC generation does not occur in nuage. Rather, this pathway is initiated when nascent, piRNA-free Siwi binds to Papi located on the mitochondrial surface. This molecular engagement depends on the Tudor domain of Papi and the symmetrical dimethylarginine residues of Siwi[8]. Siwi then binds the piRNA precursor in a stoichiometric manner and becomes the piRISC precursor (pre-piRISC). Zucchini (Zuc), an endonuclease localized on the mitochondrial surface, then processes the 3′ end of piRNA precursor within the pre-piRISC, releasing mature Siwi-piRISC from Papi to the cytosolic environment for the repression of transposons[14].

Papi contains two K-homology (KH) domains in addition to the Tudor domain[8,15]. The KH domain is widely known to confer RNA-binding activity to host proteins[16,17]. Indeed, KH domain mutants of Papi exhibited no RNA-binding activity and failed to assist Zuc-dependent piRISC maturation[14]. Thus, the RNA-binding activity of Papi via the KH domains is essential for Siwi-piRISC biogenesis. Remarkably, dephosphorylation treatment of Papi inactivated its RNA binding even though the KH domains were intact, indicating that phosphorylation is important for the regulation of the RNA-binding activity of Papi[14]. However, how the phosphorylation of Papi takes place in vivo and how it regulates the RNA-binding activity of Papi necessary for Siwi-piRISC biogenesis remains unclear.

## Results

### Ser547 phosphorylation is the key to the RNA-binding activity of Papi necessary for Siwi-piRISC biogenesis.
First, we attempted to identify the phosphorylation site(s) that influences the RNA-binding activity of Papi. Our previous UV crosslinking immunoprecipitation (CLIP), which showed the importance of Papi phosphorylation for its RNA-binding activity, was carried out in buffer without phosphatase inhibitor[14]. In this study, we

conducted CLIP with and without phosphatase inhibitor and compared the outcomes. With the inhibitor, the topmost band on the western blot was more prominent and the RNA-binding activity of the band was also stronger compared with the results without the inhibitor (Fig. 1a). We therefore isolated the topmost band (with the inhibitor) (Fig. 1b) and performed a mass spectrometric analysis. We found that Papi was phosphorylated at three residues, Ser157, Ser547, and Ser565 (Supplementary Fig. 1a–c). Ser157 is located in the second KH domain, and the two other residues are in the C-terminal auxiliary domain (Fig. 1c).

To understand which of these three residues was involved in the RNA-binding activity of Papi, we changed each of the three serine residues to alanine and expressed each of the mutants individually in BmN4 cells. The band pattern of the S157A mutant on the western blot looked similar to that of wild-type (WT) Papi (Fig. 1d). In contrast, the band patterns of the S547A and S565A mutants were slightly different from that of the WT; in particular, the top band was weaker (see the region indicated by a black triangle in Fig. 1d). The CLIP showed that the S547A mutant bound RNA only weakly, whereas, for the S157A and S565A mutants, the RNA-binding activity was comparable to that of WT Papi (Fig. 1d). This finding suggests that Ser547, but not Ser157 and Ser565, is involved in the RNA-binding activity of Papi. It is worth noting that the replacement of Ser157 in the second KH domain of Papi with phosphorylation-defective alanine did not affect the RNA-binding ability of Papi, contrary to our expectation: change(s) in charge often affects the RNA binding capacity. The S547A mutant appeared as multiple bands on the western blot (Fig. 1d), suggesting that Ser157 and Ser565 may be phosphorylated in the S547A mutant, as they are in the WT.

The S547A mutant maintained its ability to bind Siwi (Supplementary Fig. 1d). However, when this mutant was expressed alternative to endogenous Papi, the amount of Siwi-piRISC detected was negligible (Fig. 1e and Supplementary Fig. 1e). These results indicate that Siwi binding to Papi is not sufficient, but Papi phosphorylation at Ser547 is required, for Siwi-piRISC biogenesis.

### Par-1 kinase is responsible for the phosphorylation of Ser547 in Papi.
To understand whether Papi phosphorylation occurs before or after its mitochondrial targeting, we produced the Papi mutant ΔMLS, which lacks the N-terminal mitochondrial localization signal (MLS), and examined its phosphorylation status. The ΔMLS mutant was not present in the mitochondrial fraction (Fig. 2a and Supplementary Fig. 2a) and was only weakly phosphorylated (Fig. 2b and Supplementary Fig. 2b). This result suggests that Papi phosphorylation takes place on the mitochondrial surface and that the enzyme responsible for the phosphorylation of Papi is nearby.

Then, we incubated Flag-tagged Papi (Papi-Flag) in BmN4 mitochondrial lysates in the presence of $^{32}P$-γ-ATP. First, the Papi-Flag was immunopurified from BmN4 cells under harsh conditions that could theoretically avoid contamination of Papi-binding proteins and was dephosphorylated. After the incubation, Papi-Flag was $^{32}P$-labeled (Fig. 2c), suggesting the presence of kinase(s) for Papi in the fraction. Next, Papi-Flag was immunoprecipitated under ATP-free conditions and Papi-binding proteins were visualized by silver staining (Fig. 2d). Besides Papi-Flag, a number of other proteins were observed. The proteins were then eluted from the beads with a high-salt buffer and used for LC–MS/MS. A total of 649 proteins were identified, and six of them were annotated as kinases with peptide-spectrum matches greater than or equal to 2 (Fig. 2e).

Each of these six kinases was depleted by RNA interference (RNAi) and their effects on the phosphorylation of Papi were

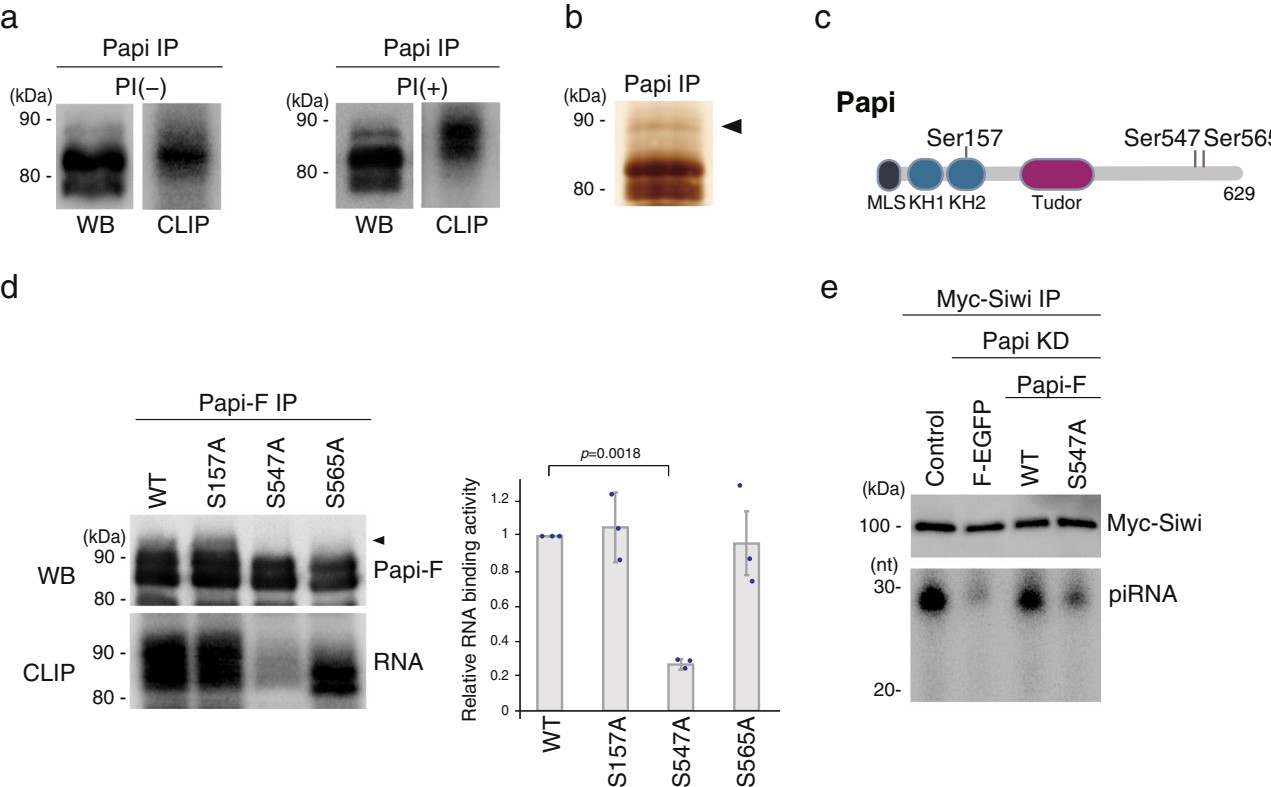

**Fig. 1 Phosphorylation of Ser547 of Papi is necessary for Siwi-piRISC biogenesis. a** Western blotting (WB) panels show band patterns of Papi in CLIP in the presence (+) and absence (−) of phosphatase inhibitor (PI). CLIP panels show the Papi-RNA complex labeled with $^{32}$P ($n = 3$). **b** Silver staining shows Papi immunoisolated from BmN4 cell lysates. The band that was used in the mass spectrometric analysis is indicated with a triangle ($n = 3$). **c** Schematic drawing of Papi shows where Ser157, Ser547, and Ser565 are located. MLS mitochondrial localization signal, KH1 and KH2 K-homology domains. **d** CLIP panel shows that the RNA-binding activity of the S547A mutant was negligible, whereas that of other mutants was comparable to that of WT Papi. The topmost band in WT Papi is indicated with a triangle. Western blotting (WB) shows the band patterns of WT Papi and its mutants in CLIP. The statistical data of the CLIP signals ($n = 3$) is shown (right). The signal intensities were calculated using ImageJ (National Institutes of Health) Each dot represents the intensity calculated from three independent experiments. *P* values were calculated by *t*-test (two-sided). Data are presented as mean values ± SD. **e** WT Papi but not the S547A mutant restored Siwi-piRISC biogenesis in cells that lacked Papi [Papi knockdown (KD)] ($n = 3$). The Papi-Flag (Papi-F) used in this experiment was RNAi-resistant. Control, Myc-Siwi was expressed in normal BmN4 cells; Flag-EGFP (F-EGFP) was used as a negative control. piRNAs were visualized by $^{32}$P-labeling. The amounts of Myc-Siwi in Input and the knockdown efficiency for Papi are shown in Supplementary Fig. 1e. Source data are provided as a Source Data file.

examined. The loss of Partitioning defective 1 (Par-1), but not of the other kinases, affected the phosphorylation of Papi (Fig. 2f and Supplementary Fig. 2c) (see the region indicated by a black triangle in Fig. 2f). Par-1 is a serine/threonine kinase (also called a MARK kinase) that functions in controlling the asymmetric distribution of factors in polarized cells, neuronal differentiation, and epithelial organization[18–20].

We next examined how the loss of Par-1 affected the phosphorylation of Ser547 in Papi. To this end, we produced a monoclonal antibody specific for Ser547 phosphorylation (Papi-pSer547) (Supplementary Fig. 2d, e). Papi-pSer547 was detected before but not after Par-1 depletion in BmN4 cells (Fig. 2g). This finding strongly supports the idea that Par-1 is involved in the phosphorylation of Ser547 in Papi. Papi appeared as multiple bands in Par-1-depleted cells (Fig. 2f), implying that Par-1 may not be involved in the phosphorylation of other Papi residues. The loss of Par-1 impaired Siwi-piRISC biogenesis (Fig. 2h), corroborating the importance of Par-1 in Siwi-piRISC production.

Because the anti-Papi-pSer547 monoclonal antibody was available, WT Papi and Papi S157A, S547A, and S565A mutants (Fig. 1d) were probed with the antibody. The results confirmed that the S157A and S565A mutants, but not the S547A mutant, were phosphorylated at Ser547, similar to the WT (Supplementary Fig. 2f). Notably, the

S565A mutant migrated faster on the gel than WT Papi. This finding explains why the topmost band of the S565A mutant looked weak on the CLIP panel (Fig. 1d), even though its RNA-binding activity was maintained similarly to that of WT (Fig. 1d). We also examined the Ser547 phosphorylation status of Papi-Flag expressed in BmN4 cells lacking Par-1 and other kinases (Fig. 2f). The anti-Papi-pSer547 monoclonal antibody did not detect Papi-Flag in Par-1-deficient cells, but did detect Papi-Flag in the other cells (Supplementary Fig. 2g). These results confirm that Par-1, and not the other kinase candidates, was responsible for the phosphorylation of Ser547 in Papi in BmN4 cells.

**Siwi targets Par-1 to mitochondrial Papi to promote its phosphorylation.** We immunoisolated both WT Papi and the ΔMLS mutant from BmN4 cells and probed the materials with the antibody for Papi-pSer547. WT Papi was detected but the ΔMLS mutant was not (Fig. 3a), which indicates that phosphorylation of Ser547 in Papi occurred only when Papi was located on the mitochondrial surface. However, Flag-Par-1 was mostly cytoplasmic (Fig. 3b). We hypothesized that an as yet unknown protein targets Par-1 to mitochondrial Papi to promote phosphorylation. One such candidate was Siwi. To test this hypothesis, we immunoprecipitated

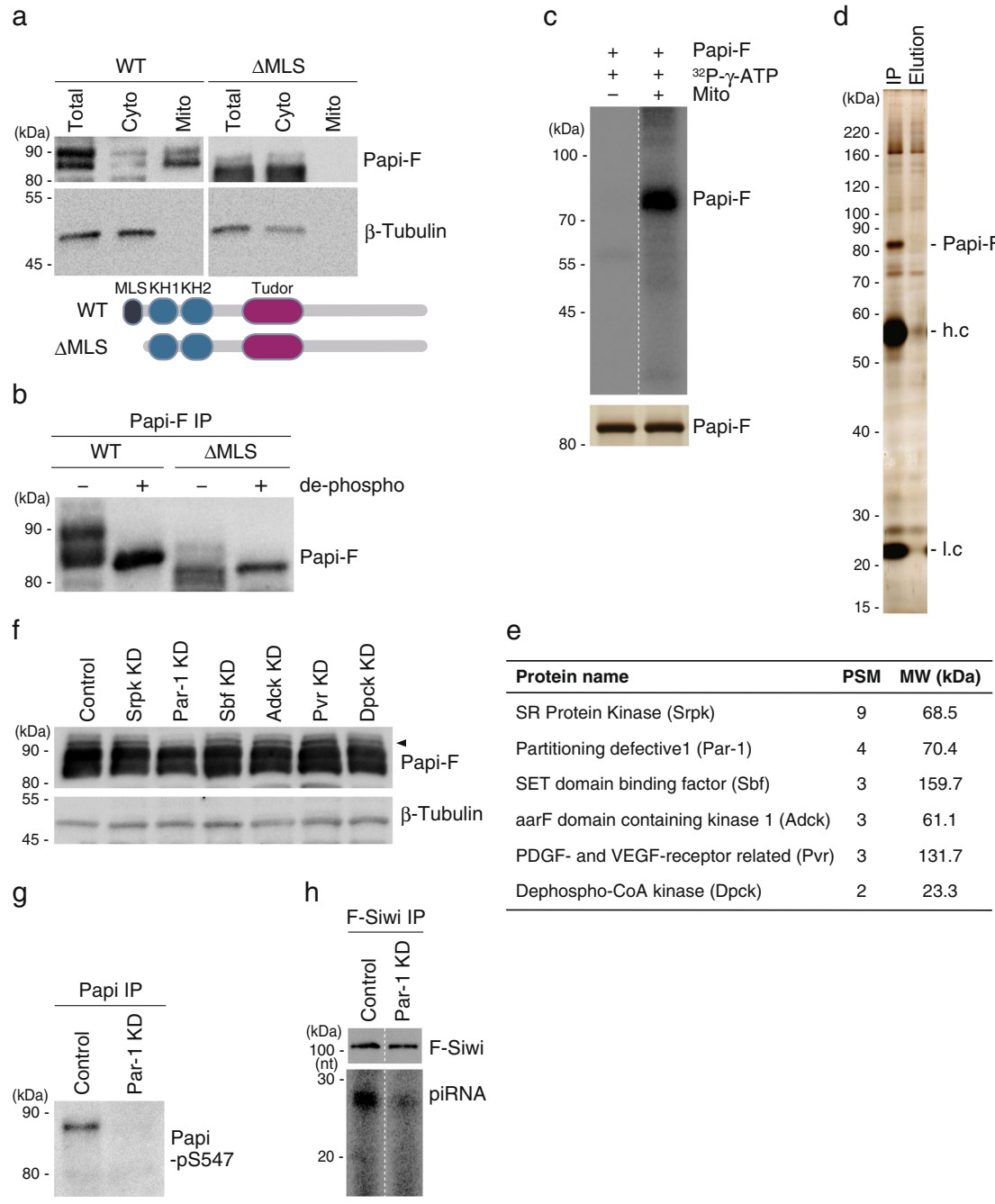

**Fig. 2 Par-1 kinase is responsible for Papi Ser547 phosphorylation. a** Western blotting shows that WT Papi is concentrated in the mitochondrial fraction (Mito), whereas the Papi ΔMLS mutant, which lacks the mitochondrial localization signal, is in the cytoplasmic fraction (Cyto) ($n = 3$). Total, total lysates of BmN4 cells. β-Tubulin was used as a marker of the cytoplasmic fraction. The schematic drawing shows the domain structure of WT Papi and the Papi ΔMLS mutant. **b** Western blotting shows that WT Papi is efficiently phosphorylated in vivo, whereas the Papi ΔMLS mutant is not ($n = 3$). de-phospho, the immunoprecipitated materials were treated with (+) and without (−) phosphatase prior to western blotting. **c** Papi-Flag (Papi-F) (upper) was $^{32}$P-labeled efficiently upon incubation with the mitochondrial fraction (Mito) of BmN4 cells in the presence of $^{32}$P-γ-ATP. Silver-stained Papi-F (lower) used in the assay is shown ($n = 3$). **d** Silver staining shows proteins that co-immunoprecipitated with Papi-F after incubation with the mitochondrial fraction of BmN4 cells. IP, proteins co-immunoprecipitated with Papi-F; Elution, proteins bound to Papi-F were eluted with high-salt buffer; h.c., heavy chain of the antibody; l.c., light chain of the antibody ($n = 3$). **e** Six kinases identified in the mass spectrometric analysis. PSM, peptide-spectrum match; MW, predicted molecular weight. **f** The six kinases in (**e**) were depleted by RNAi (KD) in BmN4 cells, and the levels of Papi-F within the cells were examined by western blotting ($n = 3$). The RNAi efficiency was examined by RT-qPCR (Supplementary Fig. 2c). β-Tubulin was used as a loading control. **g** Papi-pSer547 was detected before Par-1 depletion (Control) but not after Par-1 depletion (Par-1 KD) ($n = 3$). Anti-Papi-pSer547 antibody (Supplementary Fig. 2d, e) was used. **h** Siwi-piRISC was barely detected in Par-1-depleted cells (Par-1 KD) ($n = 3$). F-Siwi, Flag-Siwi. piRNAs were visualized by $^{32}$P-labeling. Source data are provided as a Source Data file.

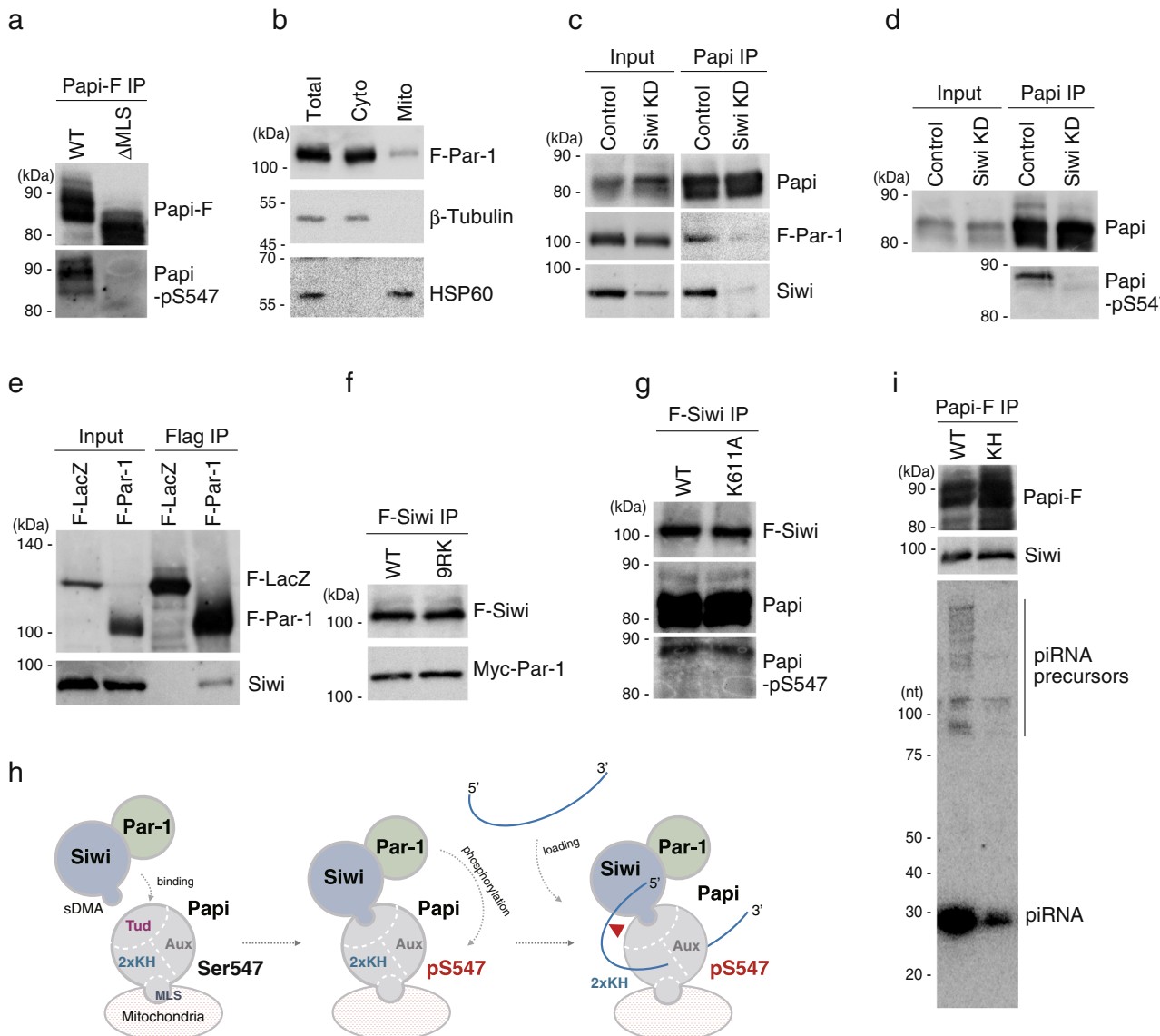

**Fig. 3 Siwi targets cytoplasmic Par-1 kinase to mitochondrial Papi for its phosphorylation. a** Western blotting shows that Ser547 of WT Papi is phosphorylated, whereas Ser547 of the Papi ΔMLS mutant is not ($n = 3$). Anti-Papi (upper) and anti-Papi-pSer547 (lower) antibodies were used. Immunoprecipitation (IP) was performed using anti-Flag antibody. **b** Western blotting shows that Flag-Par-1 (F-Par-1) is concentrated in cytoplasmic (Cyto) but not in mitochondrial (Mito) fraction ($n = 3$). β-Tubulin and HSP60 were used as markers of the cytoplasmic fraction and the mitochondrial fraction, respectively. **c** Flag-Par-1 (F-Par-1) and Siwi co-immunoprecipitated with Papi from normal BmN4 cell lysates (Control) but not from Siwi-lacking lysates (Siwi KD) ($n = 3$). The cell lysates contained both cytoplasmic and mitochondrial materials. **d** Papi immunoisolated from normal BmN4 cell lysates (Control) was phosphorylated at Ser547, whereas Papi immunoisolated from Siwi-lacking lysates (Siwi KD) was not ($n = 3$). Anti-Papi (upper) and anti-Papi-pSer547 (lower) antibodies were used. **e** Flag-Par-1 (F-Par-1) interacted with Siwi in BmN4 cytoplasmic lysates ($n = 3$). The lysates did not contain mitochondrial materials. Flag-LacZ (F-LacZ) was used as a negative control. **f** Immunoprecipitation and western blotting show that the Siwi 9RK mutant binds to Par-1 similarly to WT Siwi ($n = 3$). **g** Papi interacting with WT Siwi and the Siwi K611A mutant on the mitochondrial surface was similarly phosphorylated at Ser547 ($n = 3$). Immunoprecipitation was conducted from the mitochondrial fraction. **h** Proposed model showing that Siwi binds Par-1 kinase in the cytosol and targets it to mitochondrial Papi. Papi binds symmetrical dimethylarginine-modified Siwi via the Tudor domain. Par-1 then phosphorylates Ser547 of Papi. Papi then binds Siwi-loaded piRNA precursor via the KH domains for Zuc-mediated piRISC maturation (red triangle). The Siwi 9RK mutant cannot bind to Papi and hardly bound to piRNA precursor[14]. Thus, Siwi was loaded with piRNA precursor after its anchoring onto Papi. **i** Western blotting (upper two) shows that endogenous Siwi bound the KH mutant of Papi as it does to WT Papi ($n = 3$). Northern blotting (lowest) shows that piRNA precursors and piRNAs were barely found in the Papi KH mutant-Siwi complex (also see Supplementary Fig. 3). Source data are provided as a Source Data file.

the Papi complex from BmN4 lysates that contained Flag-Par-1 and probed the components with anti-Siwi and anti-Flag antibodies. We found that Papi interacted with Par-1 and Siwi simultaneously (Fig. 3c); however, when Siwi was depleted, the Papi-Par-1 interaction was severely weakened (Fig. 3c). This finding indicates that the interaction was Siwi-dependent. Importantly, Papi-pSer547 was

not detected in the absence of Siwi (Fig. 3d), supporting our hypothesis. Siwi was able to bind Par-1 in the cytoplasmic fraction (excluding mitochondria) (Fig. 3e). The Siwi 9RK mutant, which lacks the symmetrical dimethylarginine modification and is thus deficient in Papi binding[14], also bound to Par-1 (Fig. 3f). Together, these findings sufficiently support the model that Siwi binds Par-1 in

the cytosol, delivers it to mitochondrial Papi, and promotes its phosphorylation.

Phosphorylation of Papi was efficient even when the Siwi K611A mutant, which lacks RNA binding[21], was expressed under WT Siwi-free cell conditions (Fig. 3g). This supports the idea that Par-1-dependent phosphorylation of Papi occurs even before piRNA precursors are loaded onto Siwi but is needed for Papi to acquire RNA-binding activity. Papi then binds the Siwi-bound piRNA precursors to support Zuc-dependent piRISC maturation (Fig. 3h). Endogenous Siwi bound to the KH mutant of Papi (I69N/V142N)[22] as it does to WT Papi, but did not bind to the piRNA precursors (Fig. 3i and Supplementary Fig. 3). Under this condition, Siwi failed to become piRISC[14]. These findings show the importance of the RNA-binding activity of Papi via the KH domains to stabilize pre-Siwi-piRISC prior to Zuc-dependent piRISC maturation.

**Pseudo-phosphorylated variants of Papi can produce Siwi-piRISC but they require Siwi for their RNA-binding activity.** After the displacement of mature Siwi-piRISC, to be ready for the second-round piRISC generation, Papi should be free from any RNAs that remain bound. We hypothesized that dephosphorylation of Papi might be involved in this. To test this possibility, we changed Ser547 of Papi to aspartate or glutamate and performed rescue assays. If these mutants do not restore the generation of Siwi-piRISC abolished by the loss of endogenous Papi, this would support our hypothesis. However, both pseudo-phosphorylated variants, S547D and S547E, restored Siwi-piRISC production in Papi-depleted cells comparable to that of WT Papi (Fig. 4a and Supplementary Fig. 4a), and both variants lost the ability to bind RNAs when Siwi was depleted, similar to that of WT Papi (Fig. 4b and Supplementary Fig. 4b). This result suggests that dephosphorylation of Papi-pSer547 is unnecessary for Papi to be free from RNAs for its recycling. This also supports the intriguing concept that the function of Papi in Siwi-piRISC biogenesis is well controlled, as long as Papi has a negatively charged amino acid residue (aspartate or glutamate) near Ser547 in the auxiliary domain, but Siwi is still necessary for the Papi mutants to bind piRNA precursors.

**Cooperative control of Papi by Siwi and Par-1 is required for Papi to bind piRNA precursors.** We hypothesized that the change in surface charge of the C-terminal auxiliary domain, by phosphorylation of Papi at Ser547 or Ser547 exchange with aspartate or glutamate, alters the folding dynamics of Papi, which in turn enhances the ability of Papi to bind Siwi-bound piRNA precursors via the KH domains. The phosphorylation of Ser547 was under the cooperative control of Siwi and Par-1. We then removed the Tudor and auxiliary domains from Papi and examined the RNA-binding ability. This mutant, Papi1-222, lost the ability to bind Siwi but still exhibited the RNA-binding activity (Fig. 4c, d) in sharp contrast to WT Papi, which requires Siwi and Par-1 for its RNA-binding activity. This finding suggests that the KH domains of Papi have autonomous RNA-binding activity, but this activity is inhibited by the Tudor and auxiliary domains of Papi in vivo.

To further investigate the RNAs that were bound to the KH domains without Siwi and Par-1 control, we conducted fully automated and standardized individual nucleotide resolution CLIP (FAST-iCLIP) with WT Papi and Papi1-222 and compared the sequence reads. We mapped iCLIP-tag to piRNA precursors loaded onto Siwi upon binding to Papi on the mitochondrial surface[14]. We found that the proportion of piRNA precursors bound to Papi1-222 was much less than the proportion that bound to WT Papi (35.0% and 53.7%, respectively) (Fig. 4e). This

finding indicates that the Tudor and auxiliary domains of Papi also control the RNA-binding specificity of the KH domains.

The Papi1-480 mutant contains the KH and Tudor domains but lacks the auxiliary domain. Remarkably, this mutant bound Siwi but showed no RNA-binding activity (Fig. 4f, g). This result suggests that Siwi binding to Papi via the Tudor domain is insufficient for RNA binding, and that the auxiliary domain is required, in addition to the Tudor domain, to enhance the ability of Papi to bind Siwi-bound piRNA precursors via the KH domains. For WT Papi, the auxiliary domain has to be phosphorylated at Ser547 by Par-1 kinase. The Papi1-480 mutant was phosphatase sensitive (Supplementary Fig. 4c, d), which is reasonable because, like WT Papi, the mutant contains Ser157.

## Discussion

In this study, we successfully elucidated the functions of each of the three domains of Papi (Supplementary Fig. 5a) and the unique mechanism by which Siwi cooperates with Par-1 to control the role of Papi in Siwi-piRISC biogenesis.

The KH domains of Papi have autonomous RNA-binding activity. However, the activity of the KH domains is intramolecularly autoinhibited by the Tudor and auxiliary domains of Papi until Papi plays its functional role in Siwi-piRISC biogenesis. Siwi-piRISC biogenesis is initiated when Siwi binds to the Tudor domain of Papi via symmetrical dimethylarginine residues (Supplementary Fig. 5b). Siwi simultaneously promotes the phosphorylation of Ser547 in the auxiliary domain by targeting Par-1 kinase to Papi. This phosphorylation presumably alters the folding and/or other aspects of Papi, thereby allowing Papi to bind Siwi-bound piRNA precursors via the KH domains. Substitution of Ser547 with alanine inhibited phosphorylation, abrogated RNA binding of Papi, and attenuated Siwi-piRISC biogenesis in the BmN4 cells. Conversely, the Papi1-222 mutant that lacked the Tudor and auxiliary domains autonomously bound RNAs in vivo, but the binding was less biased toward piRNA precursors. The Papi1-222 mutant still bound to piRNA precursors, albeit weakly. This may be because piRNA precursors are concentrated near the mitochondrial surface in BmN4 cells, which is the location for Ago3-independent Siwi-piRISC biogenesis. The Papi1-480 mutant bound Siwi, but not piRNA precursors (or any other RNAs). These results indicate the importance of the auxiliary domain, which is independent of the Tudor domain.

Substitution of Ser547 with aspartate or glutamate blocked phosphorylation by Par-1, but Siwi-piRISC was produced in the presence of Siwi. However, depletion of Siwi impaired RNA binding of these mutants. This suggests that Siwi binding to Papi via the Tudor domain has an additional role, which is to provide piRNA precursors per se for binding to Papi. In this way, the generation of Siwi-piRISC capable of repressing transposons is secured.

Mouse Papi (known as Tdrkh) and *Drosophila* Papi do not have a serine residue at the site that corresponds to Ser547 in *Bombyx* Papi (Supplementary Fig. 5c). The auxiliary domain of Tdrkh is much shorter than that of *Bombyx* Papi, and the peptide sequence of the auxiliary domain is less conserved than the rest of the sequence. DISOPRED3 predicted that Tdrkh and Papi have two intrinsic disordered regions, N-IDR and C-IDR (Supplementary Fig. 5d). In the C-IDR of Papi (Pro463 to the end; 164 resides), which harbors Ser547, 14.6% of the amino acids are aspartate or glutamate. In the corresponding region of Tdrkh (Asp497 to the end; 63 residues), 27.0% of the amino acids are aspartate or glutamate. Therefore, even though phosphorylation of Tdrkh is similar to that of Papi, the addition of a phosphate group to the region would not strongly modulate the charge of

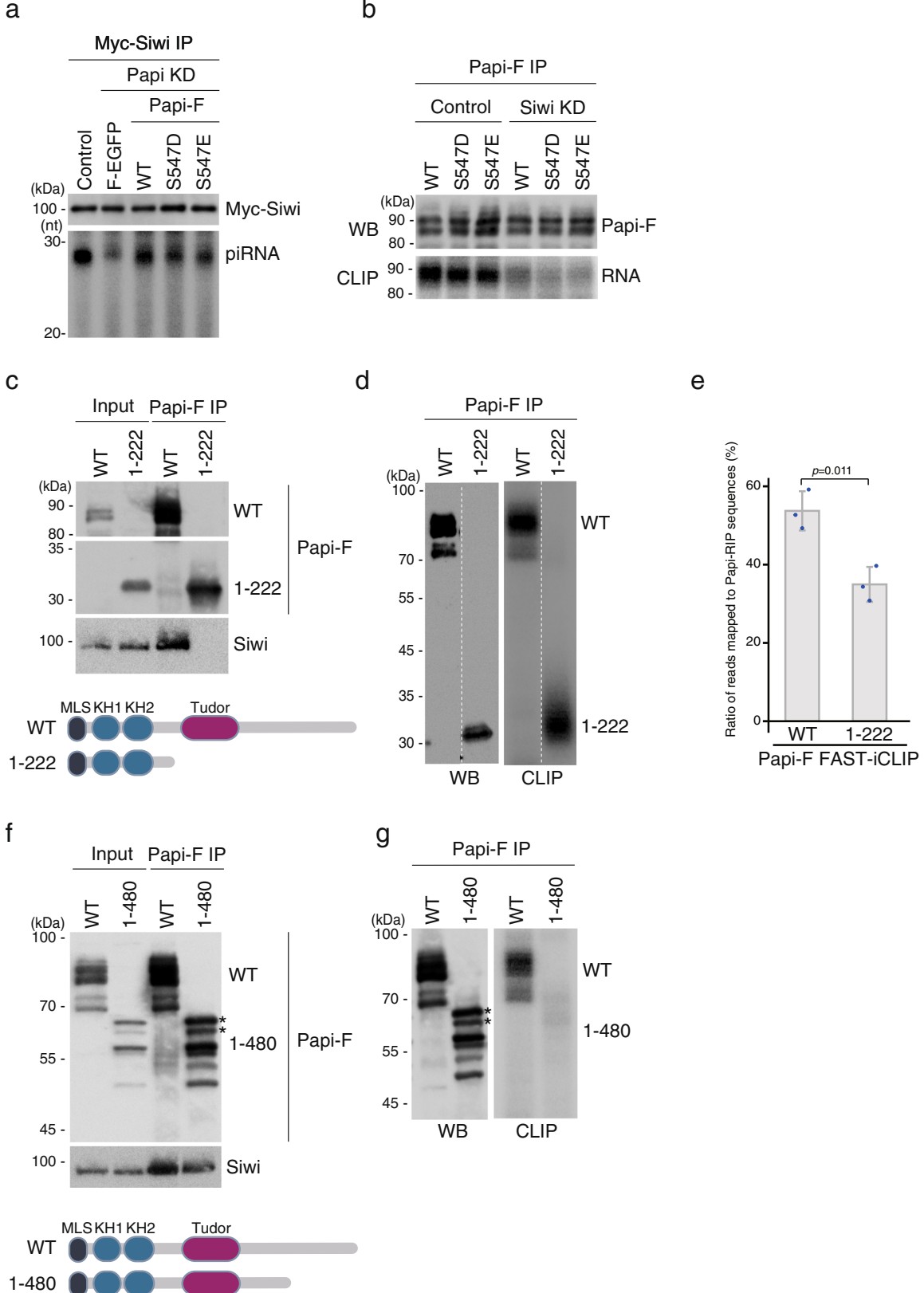

the domain. The C-IDR of Tdrkh (Glu525 to the end, 35 residues) is much shorter than that of Papi, and therefore the auxiliary domain of Tdrkh may be unable to mimic the functions of the auxiliary domain of Papi.

In *Drosophila* ovarian somatic cells, Papi is dispensable for the production of piRISC[23,24]. Recent studies have shown that a heterodimer consisting of Gasz and Daedalus acts as a scaffold for piRISC biogenesis[25,26]. Gasz and Daedalus, like Papi, are localized on the mitochondrial surface, but lack known RNA-binding domains such as the KH domain. This indicates that the functional contribution of Gasz and Daedalus to piRISC biogenesis, other than scaffolding, may be different from that of Papi. These

**Fig. 4 Auxiliary domain of Papi influences the RNA-binding activity of Papi via the KH domains. a** Papi S547D and S547E variants restored the Siwi-piRISC production in Papi-depleted cells (Papi KD) ($n = 3$). All Papi-Flag (Papi-F) employed in this experiment was RNAi-resistant. Control, Myc-Siwi was expressed in normal BmN4 cells; Flag-EGFP (F-EGFP) was used as a negative control. piRNAs were visualized by $^{32}$P-labeling. The amounts of Myc-Siwi in Input and the knockdown efficiency for Papi are shown in Supplementary Fig. 4a. **b** CLIP shows that the RNA-binding activity of the S547D and S547E variants was reduced significantly following Siwi depletion (Siwi KD), which is similar to that of the WT ($n = 3$). Western blotting (WB) shows the band patterns of WT Papi and its mutants in CLIP. **c** Immunoprecipitation and western blotting show that, unlike WT Papi, the Papi1-222 mutant did not bind Siwi ($n = 3$). The schematic drawing shows the domain structure of WT Papi and Papi1-222. **d** CLIP shows that the Papi1-222 mutant binds RNAs ($n = 3$). Western blotting (WB) shows the band patterns of WT Papi and the mutant in CLIP. **e** Bar graph shows the mean ratio of reads of Papi-Flag FAST-iCLIP tags that mapped to Papi-RIP sequences ($n = 3$). Each dot represents the ratio calculated from three independent experiments. $P$ values were calculated by $t$-test (two-sided). Data are presented as mean values ± SD. **f** Immunoprecipitation and western blotting show that the Papi1-480 mutant binds Siwi as well as WT Papi ($n = 3$). Asterisks indicate phosphorylated Papi1-480 (see Supplementary Fig. 4c, d). Schematic drawing shows the domain structure of WT Papi and Papi1-480. **g** CLIP shows that the Papi1-480 mutant did not bind RNAs ($n = 3$). Western blotting (WB) shows band patterns of WT Papi and the Papi1-480 mutant in CLIP. Asterisks, phosphorylated Papi1-480. Source data are provided as a Source Data file.

findings are emblematic of the interspecies diversity of the piRNA pathway, even though the purpose of the mechanism is the same.

How does Papi initiate the next reaction? We first considered that Papi was recycled by dephosphorylation. This would mean that any residual RNAs left on Papi after the Zuc reaction would be instantly released from Papi. However, this possibility was ruled out because the pseudo-phosphorylated variants functioned similar to WT Papi. Next, we considered the possibility that Papi was degraded immediately after the first round of piRISC formation, and the unphosphorylated nascent Papi functions in the second round. Another possibility we considered was that the residual RNAs left on Papi in the first round are simply expelled by unloaded Siwi, which binds to Papi for the next round. We favor the former scenario because Papi with phosphorylated Ser547 was rare in BmN4 cells, as was repeatedly shown in this study.

This is an important study to show how the quality of piRISC, which determines the degree of transposon repression in vivo, is highly regulated by post-translational modification of piRISC biosynthetic factors and that this regulation is driven by PIWI.

A consensus sequence for Par-1 to mediate phosphorylation was reported in mammals[27], but we did not find a consensus-like sequence in silkworm Papi. We concluded that, although a consensus may be present for mammalian Par-1 substrates, this may not be the case for *Bombyx* substrates.

In *Drosophila*, the substrates for Par-1 include Tau[28], Myosin-II (Myo-II)[29], Mind bomb (Mib)[30], and Hippo signaling (Hpo)[31]. Tau and Myo-II are involved in the regulation of microtubule dynamics, and Mib and Hpo are involved in the regulation of neurogenesis and tissue growth, respectively. In the germline, Oskar and Bazooka/Par-3 are well known substrates for Par-1. Both these proteins regulate the establishment of the anterior/posterior axis[32,33], but the effect of Par-1 phosphorylation on each protein is different; that is, Par-1 regulates the stability of Oskar and the subcellular localization of Bazooka/Par-3. Although Papi, Oskar, and Bazooka/Par-3 are all phosphorylated by Par-1 in the germline, the present study shows that Par-1 regulates the RNA-binding activity of a specific RNA-binding protein, namely Papi.

## Methods

**Cell culture.** BmN4 cells were cultured at 26 °C in EX-CELL® 420 Serum-Free Medium for insect cells (Sigma) supplemented with 10% fetal bovine serum (Equitech-Bio) and penicillin-streptomycin-glutamine (Thermo Fisher Scientific).

**RNAi and transgene expression.** A total of $1 \times 10^6$ BmN4 cells and 500 pmol of short interfering RNA (siRNA) duplex were suspended in 100 μL of EP buffer [137 mM sodium chloride, 5 mM potassium chloride, 0.5 mM sodium hydrogen phosphate, 2.1 mM HEPES-KOH (pH 7.1)] and electroporation was conducted using a Nucleofector 2b device with program T-001 (Lonza Bioscience). This procedure was repeated twice. After 12 days, the knockdown cells were harvested.

The siRNA sequences used are shown in Supplementary Table S1. Double-stranded RNAs (dsRNAs) were produced by in vitro T7 transcription, followed by annealing in water. The PCR primers used to generate dsRNAs are summarized in Supplementary Table S1. BmN4 cells were transfected with 2 μg of dsRNAs (per $4 \times 10^5$ cells) using FuGENE HD (Promega). This procedure was repeated three times. After 9 days, the transfected cells were harvested. To express proteins exogenously in BmN4 cells, the cells were transfected with 2 μg of plasmids (per $6 \times 10^5$ cells) using FuGENE HD. After 48 h, the transfected cells were harvested.

**Cloning of *Par-1* cDNA.** *Par-1* cDNAs were obtained by RT-qPCR using total RNA that was extracted from *Bombyx* ovaries. The PCR primers used were summarized in Supplementary Table S1.

**Plasmid construction.** Plasmids pIB-3×Myc and pIB-3×Flag were generated using a pIB vector (Thermo Fisher Scientific). Vectors to express Flag-Par-1 were generated by inserting *Par-1* cDNAs into pIB-3×Flag using NEBuilder HiFi DNA Assembly Master Mix (New England Biolabs). A DNA fragment encoding β-galactosidase was PCR-amplified from pMT/V5-His/lacZ (Thermo Fisher Scientific) and cloned into the PCR amplified pIB-3×Flag vector. Then, the expression vector of F-LacZ was generated using NEBuilder HiFi DNA Assembly Master Mix. Vectors for expressing the Papi-Flag mutants (S157A, S547A, S565A, ΔMLS, S547D, S547E, 1-222, and 1-480) were generated by inverse PCR, using RNAi-resistant Papi-Flag[14] as the template. A DNA fragment encoding Papi-Flag WT and 1-480 were PCR-amplified from pIB-Papi-3×Flag and pIB-Papi-1-480-3×Flag respectively and cloned into the PCR amplified pET-28a(+) vector (Sigma) Then, the vectors of Papi-F WT and 1-480 for TnT® was generated using NEBuilder HiFi DNA Assembly Master Mix. The PCR primers used are summarized in Supplementary Table S1.

**Production of an anti-Papi-pS547 monoclonal antibody.** To generate the anti-Papi-pS547 monoclonal antibody, synthetic peptides corresponding to the amino acids in positions 540-555 of Papi [DRHPLSI(pS)NFDLSYP] that was conjugated with keyhole limpet hemocyanin were purchased from Eurofin Genomics. The peptides were immunized into BALB/C mice. Three days after the second boost, the lymph node cells were fused with myeloma cell line Sp2 using an ECFG Super Electro Cell Fusion Generator device (Nepa Gene). The culture media were screened by ELISA and following western blotting.

**Western blotting.** Proteins separated on sodium dodecyl sulfate (SDS) polyacrylamide gels were blotted on polyvinylidene di-fluoride membranes. The membranes were blocked with 5% PhosphoBlocker Blocking Reagent (Cell Biolabs) or 5% skimmed milk in phosphate-buffered saline (PBS; FUJIFILM Wako Pure Chemical) followed by incubation with antibodies diluted in 0.1% Tween-20 in PBS (T-PBS). The membranes were extensively washed with T-PBS after each procedure. The primary antibodies used in this study were anti-Papi monoclonal antibody (1:1000 dilution)[14], anti-Siwi monoclonal antibody (1:5000 dilution)[6], anti-Papi-pS547 monoclonal antibody (supernatant of hybridoma cells; this study), anti-Flag M2 monoclonal antibody (1:1000 dilution; Sigma), anti-Flag antibody produced in rabbit (1:1000 dilution; Sigma), anti-Myc monoclonal antibody (1:1000 dilution; 9E10, Developmental Studies Hybridoma Bank), anti-β-Tubulin monoclonal antibody (1:1000 dilution; E7, Developmental Studies Hybridoma Bank), and anti-HSP60 monoclonal antibody (1:1000 dilution; LK1, StressMarq Biosciences). Peroxidase-conjugated anti-mouse IgG (1:5000 dilution; Cappel) and anti-rabbit IgG (1:1000 dilution; Cell Signaling Technology) antibodies were used as secondary antibodies. The membranes were incubated with Clarity Western ECL Substrate (Bio-Rad), and images were collected using a ChemiDoc XRS Plus System (Bio-Rad).

**Immunoprecipitation**. BmN4 whole and mitochondrial lysates were prepared in binding buffer [30 mM HEPES (pH 7.3), 150 mM potassium acetate, 5 mM magnesium acetate, 5 mM dithiothreitol (DTT), 0.5%Triton X-100, 2 μg/mL pepstatin, 2 μg/mL leupeptin, and 0.5% aprotinin] containing Phosphatase Inhibitor Cocktail Solution I (FUJIFILM Wako Pure Chemicals) and incubated with anti-Papi or anti-Flag antibody bound to Dynabeads Protein G (Thermo Fisher Scientific) at 4 °C for 2 h. The beads were washed 3 times with binding buffer. Eluted proteins were separated by SDS-PAGE and detected by western blotting (see Western blotting) or silver staining using a SilverQuest Silver Staining Kit (Thermo Fisher Scientific). RNAs were eluted from the beads by phenol–chloroform after proteinase K treatment and precipitated with ethanol. RNA $^{32}$P-labeling was carried out using T4 polynucleotide kinase (New England Biolabs).

**Dephosphorylation treatment**. Immunopurified Papi and Papi-Flag from BmN4 cells were incubated with Lambda Protein Phosphatase (New England Biolabs) at 30 °C for 30 min for dephosphorylation.

**Rescue assay**. A total of $1 \times 10^6$ BmN4 cells and 500 pmol of the siRNA duplex were suspended in 100 μL of EP buffer and electroporation was conducted using a Nucleofector 2b device with program T-001. Then, 2 days after electroporation, the cells were transfected with 500 pmol of the siRNA duplex using FuGENE HD. The siRNA sequences are presented in Supplementary Table S1. After RNAi, the cells were transfected with 2 μg of Papi-Flag plasmid using FuGENE HD and incubated at 26 °C for 48 h. The cells were again transfected with 500 pmol of the siRNA duplex using FuGENE HD, and then transfected with 2 μg of Myc-Siwi plasmid and incubated at 26 °C for 24 h. BmN4 lysates were prepared in binding buffer [30 mM HEPES (pH 7.4), 150 mM potassium acetate, 5 mM magnesium acetate, 5 mM DTT, 0.1% Tergitol-type NP-40, 2 μg/mL pepstatin, 2 μg/mL leupeptin, 0.5% aprotinin], and incubated with anti-Myc antibody bound to Dynabeads Protein G at 4 °C for 2 h. The beads were washed twice with binding buffer containing 500 mM sodium chloride and then twice with binding buffer. RNAs were eluted from the beads by phenol–chloroform after proteinase K treatment for 20 min at 37 °C and precipitated with ethanol. RNA radiolabeling was carried out as described (see Immunoprecipitation).

**Northern blotting**. RNAs separated on 10% acrylamide-denaturing gels were blotted on a Hybond-N+ membrane (Cytiva). After UV crosslinking, the membrane was incubated at 42 °C for 30 min in hybridization buffer [200 mM sodium phosphate buffer (pH 7.2), 7% SDS, 1 mM ethylenediaminetetraacetic acid (EDTA)] as a pre-hybridization treatment. Oligonucleotide probes were $^{32}$P-labeled using T4 polynucleotide kinase and hybridized to the membrane in a hybridization buffer at 42 °C overnight. The membranes were washed in saline sodium citrate buffer [30 mM sodium citrate (pH 7.0), 300 mM NaCl] supplemented with 0.1% SDS at 42 °C, and autoradiographs were obtained using Typhoon FLA 9500 (Cytiva). The probe sequences are shown in Supplementary Table S1.

**Mitochondrial fraction preparation**. A total of $5 \times 10^6$ cells were suspended with 3 mL of buffer for first resuspension [30 mM Tris-HCl (pH 7.4), 225 mM D-mannitol, 75 mM sucrose, 0.05 mM EGTA] and ruptured by passing five times through a 25-gauge needle and ten times through a 30-gauge needle attached to a 1 mL syringe. The lysates were centrifuged at $600 \times g$ for 5 min, and the pellets of nucleus and cell debris were discarded. This step was repeated twice. The lysates were centrifuged at $7000 \times g$ for 10 min, and the supernatants were collected as the cytoplasmic fraction. The pellets were collected and resuspended with 3 mL of buffer for the second resuspension [30 mM Tris-HCl (pH 7.4), 225 mM D-mannitol, 75 mM sucrose]. This step was repeated at $10,000 \times g$ for 10 min. The final pellets were collected as the mitochondrial fraction. Proteins in each fraction were detected by western blotting.

**In vitro phosphorylation assay**. BmN4 whole cell lysate was prepared in binding buffer [30 mM HEPES (pH 7.3), 150 mM potassium acetate, 5 mM magnesium acetate, 5 mM DTT, 0.5% Triton X-100, 2 μg/mL pepstatin, 2 μg/mL leupeptin, 0.5% aprotinin] and incubated with anti-Flag M2 antibody bound to Dynabeads Protein G at 4 °C for 1 h. The beads were washed twice with binding buffer containing 500 mM sodium chloride and then twice with binding buffer, followed by dephosphorylation treatment (see Dephosphorylation treatment). Beads binding with Papi-F were incubated with the BmN4 mitochondrial fraction (see Mitochondrial fraction preparation) at 4 °C for 2 h. Beads were washed three times with binding buffer. Beads were incubated in vitro phosphorylation buffer [50 mM HEPES (pH 7.3), 150 mM sodium chloride, 0.1 mM EGTA, 10 mM magnesium chloride, 10% glycerol, 1 mM DTT, Phosphatase Inhibitor Cocktail Solution I] at 26 °C for 15 min[34]. The beads were washed twice with binding buffer and the proteins were eluted. After SDS-PAGE, the proteins were visualized by silver staining using a SilverQuest Silver Staining Kit, and $^{32}$P radiolabeled proteins were detected using a Typhoon FLA 9500.

**Protein identification by LC-MS/MS**. The destained gel pieces were incubated with 10 mM DTT in 100 mM ammonium bicarbonate for 30 min at 60 °C for protein reduction, and subsequently incubated with 55 mM iodoacetamide for 1 h at room temperature with shielding from light for alkylation. Alkylated proteins in the gel pieces were digested with 10 ng/μL of Trypsin Gold (Promega) at 37 °C for 18 h. Supernatants were transferred to new tubes. Peptides that remained in the gel pieces were extracted by shaking with ultrapure water for 20 min, 25% acetonitrile (ACN), and 0.1% trifluoroacetic acid (TFA) for 20 min, 50% ACN and 0.1% TFA for 20 min, 99.9% ACN and 0.1% TFA until the gels were completely shrunk. Collected supernatants were dried up to 10 μL. The liquid chromatography-tandem mass spectrometry (LC–MS/MS) analysis was conducted using an LTQ-Orbitrap Velos mass spectrometer (Thermo Fisher Scientific) equipped with a Zaplous Advance nano UHPLC HTS-PAL xt System (AMR) containing an autosampler and nano-HPLC system. Mobile phases consisted of (A) 0.1% formic acid in ultrapure water and (B) 100% ACN. Samples were loaded onto a trap column (5 μm, 0.3 mm ID × 5 mm, L-column; CERI) and directly connected to a Zaplous α Pep-C18 packed column (3 μm, 0.1 mm × 150 mm) (AMR). The nanoLC gradient was delivered at 500 nL/min and consisted of a linear gradient of mobile phase B developed from 5% to 45% in 90 min. A spray voltage of 1.5 kV was applied. The mass spectrometer was operated in positive ionization mode. Full MS scans were performed in the Orbitrap mass analyzer. The mass spectrum was acquired over a mass range of 350–1500 Da with a resolution of 30,000 FWHM (full width half maximum). The ten most intense precursor ions were selected for each of MS/MS scans. The ten multiply charged ions were sequentially isolated and fragmented in the linear ion trap by collision-induced dissociation. For the MS/MS, the ion selection threshold was set to 1000 counts, the activation Q was set to 0.25, and the activation time was set to 10 ms. The data were searched against the protein sequence database from the Silkworm Genome Research Program (http://sgp.dna.affrc.go.jp/pubdata/genomicsequences.html) using the search program Proteome Discoverer 2.1 (Thermo Fisher Scientific) for protein identification. Spectra were searched with a mass tolerance of 10 ppm in MS mode and 0.6 Da in MS/MS mode, allowing up to two missed cleavage sites. The minimum peptide length was set as 6. Carbamidomethylated Cys was searched as a fixed modification, whereas oxidized Met and phosphorylated Ser, Thr, and Tyr were searched as variable modifications. We used the processing node "ptmRS" in the Proteome Discoverer workflow. The ptmRS score cutoff (Site Probability Threshold in PSM Grouper node) was 75. The protein FDR cutoff was 0.01 (FDR 1%), the peptide FDR cutoff was 0.05 (FDR 5%), and the modification FDR cutoff was 0.01% (FDR 1%). The confidence of peptides that contain Ser157, Ser547, and Ser565 was high for all (FDR < 0.01).

**RT-qPCR**. Total RNA was extracted from BmN4 cells using ISOGEN II (NIPPON GENE), followed by DNase treatment. Then, 1 μg of total RNAs was reverse transcribed using ReverTra Ace qPCR RT Master Mix (TOYOBO). The cDNAs were amplified with PowerUp SYBR Green Master Mix (Thermo Fisher Scientific) using StepOne-Plus (Thermo Fisher Scientific, v2.2.2). The primer sequences are shown in Supplementary Table S1. Cells were prepared by three independent knockdown experiments. Quantification of the mRNAs was carried out using the Δ ΔCt method and datasets obtained by the three independent knockdown experiments.

**In vitro translation**. Unmodified Flag-tagged Papi proteins were synthesized in vitro using a TnT® T7 Quick Coupled Transcription/Translation System (Promega). The pET-28a(+)-BmPapi-3×Flag and pET-28a(+)-BmPapi-1-480-3×Flag plasmids were used as the template for the transcription and translation.

**CLIP**. For UV crosslinking, BmN4 cells were washed once with ice-cold PBS. UV crosslinking was performed with an irradiance of 200 mJ/cm$^2$ at 254 nm. Cells were pelleted and lysed with lysis buffer [20 mM HEPES (pH 7.3), 150 mM sodium chloride, 1 mM EDTA, 1 mM DTT, 0.5% Triton X-100, 2 μg/mL pepstatin, 2 μg/mL leupeptin, 0.5% aprotinin]. RNA–protein complexes were immunoprecipitated using 3 μg of anti-Flag M2 antibody and 30 μL of Dynabeads Protein G at 4 °C for 2 h. Beads were washed three times with wash buffer [20 mM HEPES, (pH 7.3), 300 mM sodium chloride, 1 mM DTT, 0.5% Triton X-100, 2 μg/mL pepstatin, 2 μg/mL leupeptin, 0.5% aprotinin], followed by partial RNA digestion with 1 U/μL RNase T1 (Thermo Fisher Scientific) for 15 min at room temperature. Beads were washed three times with high-salt wash buffer [20 mM HEPES (pH 7.3), 500 mM sodium chloride, 1 mM DTT, 0.5% Triton X-100, 2 μg/mL pepstatin, 2 μg/mL leupeptin, 0.5% aprotinin]. The 5′ ends of the RNAs were radiolabeled with $^{32}$P in binding buffer using T4 polynucleotide kinase, followed by SDS-PAGE and detection using a Typhoon FLA 9500.

**FAST-iCLIP**. The 10-cm dish BmN4 cells transfected with Papi-Flag or Papi1-222-Flag were used for each iCLIP experiment. For UV crosslinking, the BmN4 cells were washed once with ice-cold PBS. UV crosslinking was performed with an irradiance of 200 mJ/cm$^2$ at 254 nm. The cells were pelleted and lysed with lysis buffer [20 mM HEPES (pH 7.3), 150 mM sodium chloride, 1 mM EDTA, 1 mM DTT, 0.5% Triton X-100, 2 μg/mL pepstatin, 2 μg/mL leupeptin, 0.5% aprotinin]. RNA–protein complexes were immunoprecipitated using 12 μg of anti-Flag M2 antibody and 120 μL of Dynabeads Protein G at 4 °C for 2 h. The beads were washed three times with wash buffer [20 mM HEPES (pH 7.3), 300 mM sodium chloride, 1 mM DTT, 0.5% Triton X-100, 2 μg/mL pepstatin, 2 μg/mL leupeptin, 0.5% aprotinin], followed by partial RNA digestion with 1 U/μL RNase T1 for

15 min at room temperature. Then, the beads were washed three times with high-salt wash buffer [20 mM HEPES (pH 7.3), 500 mM sodium chloride, 1 mM DTT, 0.5% Triton X-100, 2 µg/mL pepstatin, 2 µg/mL leupeptin, 0.5% aprotinin]. After 3′ end RNA dephosphorylation with T4 polynucleotide kinase, the RNAs were ligated at their 3′ ends to a preadenylated RNA adaptor with T4 RNA ligase 2, truncated KQ (New England Biolabs), and then radioactively labeled. The samples were analyzed by NuPAGE (Life Technologies) and RNA–protein complexes were transferred to nitrocellulose. After cutting out the region of nitrocellulose containing the RNA–protein complexes, RNAs were removed from the membrane by proteinase K (Roche) digestion. Samples were processed for reverse transcription, cDNA purification, and cDNA circularization[35]. Libraries were then amplified with Q5 DNA polymerase (New England Biolabs) and size-selected by PAGE. The iCLIP libraries were sequenced on the MiSeq platform as single-end 51-bp reads.

**FAST-iCLIP data analysis**. Sequencing reads were trimmed by a 3′ adapter sequence using Cutadapt[36], and reads shorter than 27 bp were discarded. PCR duplicates were removed by collapsing all identical reads containing the same random barcode using the fastx-collapser in the FASTX-Toolkit (http://hannonlab.cshl.edu/fastx_toolkit), then the 5′-end random barcode at positions 1–9 of the reads was removed. The reads were mapped against the *Bombyx mori* genome assembly (November 2016) from the SilkBase database[37] using STAR[38]. The reads that mapped to the genome were mapped against the Papi-RIP data obtained previously[14] (Gene Expression Omnibus [GEO] accession GSE107371) using Bowtie[39], allowing a single mismatch. The Papi-RIP data were downloaded from the GEO and processed with minor modification of the genome mapping. Genome mapping was performed using the same method we used for the FAST-iCLIP data, as described above. The ratio of reads mapped to Papi-RIP sequences against the reads mapped to the genome sequences was calculated and is shown as a bar graph using R.

**Reporting summary**. Further information on research design is available in the Nature Research Reporting Summary linked to this article.

## Data availability
The data supporting the findings of this study are available from the corresponding authors upon reasonable request. The FAST-iCLIP data were deposited in the Gene Expression Omnibus under accession code GSE179432. The mass spectrometry data for Papi phosphorylation sites were deposited in the proteomeXchange Consortium via the JPOST under accession code PXD027395. The mass spectrometry data for Papi interacting proteins were deposited in the proteomeXchange Consortium via the JPOST under accession code PXD027396. Source Data are provided with this paper.

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

## Acknowledgements
We thank H. Siomi, S. Yamanaka, H. Yamazaki, S. Hirakata, Y. Namba, and R. Onishi for reading the manuscript and providing useful comments. We also thank all other members of the Siomi laboratory at The University of Tokyo. This study was supported by research grants from MEXT to M.C.S. (19H05466), K.M.N. (20K06483), and Y.W.I. (21H00259). H.Y. is supported by the Japan Society for the Promotion of Science (19J13939), and Y.W.I. is supported by JST PRESTO (JPMJPR20E2).

## Author contributions

H.Y., K.M.N., and Y.I. performed the biochemical experiments. H.Y. and L.N. performed the mass spectrometry. Y.W.I. performed the bioinformatics. All authors analyzed the data and participated in writing the manuscript. M.C.S. supervised all of the research.

## Competing interests

The authors declare no competing interests.
