## [Peer Review File · Nature Communications]

Title: Siwi cooperates with Par-1 kinase to resolve the autoinhibitory effect of Papi for Siwi-piRISC biogenesisREVIEWER COMMENTS

Reviewer #1 (Remarks to the Author):

In this manuscript, the authors used MS to identify the phosphorylation sites of Papi. They found three potential phosphorylation sites and provided the MS/MS spectra of the corresponding phosphopeptides (Fig.1c). However, I am not convinced by these spectra due to the following reasons:

1. All three peptides have missed cleavage sites, if these are real phosphopeptides, the authors should also see the phosphopeptide signals without miss cleavage sites, which should have better intensities and much easier to be identified.
2. In these three MS/MS spectra, all the main/strongest peaks are not assigned, the assigned b and y fragments are relatively weak, which decrease the identification confidence.
3. In MS/MS fragmentation, cleavage at N-terminal to proline usually generates the strongest b and y ions, but we could not see that phenomenon in the spectra of the peptides bearing S157 or S547.

Taken these into consideration, although later the authors provided evidence of phosphorylation on S547, it might not be the predominate phosphorylation site(s), as we could see two major Papi bands around 80 kDa, and also multiple bands in Papi1-480, suggesting other modification sites. Moreover, radioactive assay (Fig.2c) also indicated the predominate phosphorylated Papi is around 80 kDa. The authors may also want to analyze the two bands around 80 kDa but not 90 kDa, although the CILP showing the signal around 90 kDa.

To sum up, the present form of this manuscript is not suitable for publication in a prestigious journal such as Nature Communications and I would recommend REJECTION.

Reviewer #2 (Remarks to the Author):

Bombyx Papi facilitates Siwi-piRISC biogenesis via directly binding to both Siwi protein and Siwi-loaded piRNA precursors. Phosphorylation of Papi is required for its binding to piRNA precursors and crucial to Siwi-piRISC biogenesis. In this article, the authors studied how Papi is phosphorylated and how the phosphorylation regulates the RNA-binding activity of Papi for Siwi-piRISC biogenesis. First, the authors identified residue Ser547 as the critical phosphorylation site in Papi for Siwi-piRISC biogenesis. Next, they identified Par-1, located nearby the mitochondrial surface, as the kinase for phosphorylation of Papi. Their mechanistic studies showed that cytoplasmic Siwi recruits Par-1 to phosphorylate Papi, resulting in the alteration of the folding dynamics of Papi and in turn promoting Zuc-dependent Siwi-piRISC maturation. These findings are of high interest and I support their publication in Nature

Communications. Most of the experiments are well done, and many of the data presented are convincing and support the conclusions. I only have a few comments to improve the manuscript, as follows.

1. In Fig 1d, the point mutation S157A in KH domain appears to cause a moderate reduction of RNA-binding capacity compared with wildtype Papi. In principle, removal of negative charged phosphorylation in KH domain should be in favor of RNA binding. How this mutation in the KH domain could affect the RNA-binding activity is not discussed. Also, it seems to me that the S565A mutant shows a similar reduction of phosphorylation. It's better to confirm these IB data using the Papi-pS547 monoclonal antibody.
2. In Fig 2f, the phosphorylated Papi band is fuzzy. To validate the specificity of Par-1 to phosphorylation of Ser547, it's better to re-examine the effects of six kinases on Papi phosphorylation using the Papi-pS547 monoclonal antibody.
3. In Fig. 3e and 3f, the authors concluded that Siwi binds to Par-1 in the cytoplasmic fraction before piRNA precursors are loaded onto Siwi. I am wondering whether the mature Siwi-piRISC, which has been released from Papi and localized in cytoplasm, could still interact with Par-1.
4. The labels on Western blotting panels should be improved.

Reviewer #3 (Remarks to the Author):

Dear editor,

this manuscript by Yamada et al. entitled "Siwi cooperates with Par-1 kinase to resolve the autoinhibitory effect of Papi for Siwi-piRISC biogenesis" finds that phosphorylation of Papi by Par-1 is required for piRNA biogenesis. Papi is a conserved, mitochondria-localized scaffold protein important for piRNA production that harbors KH domains, a Tudor domain important for Siwi binding, and an unstructured C-terminus. The authors show that phosphorylation of a specific residue (Ser547) within the unstructured C-terminus is relevant for the binding of RNA precursors by Papi's KH domain and affects piRNA biogenesis. The authors study several Papi mutants and hypothesize that phosphorylation by Par-1 results in a conformational change that opens up the KH domains and allows piRNA processing to occur. While this is an interesting model, I would ideally like to see more data added on how this phosphorylation mechanistically affects piRNA biogenesis. Overall, while I feel that the work in its current form contains several nice observations that would be interesting to the readership of Nature Communications, the manuscript would benefit from additional data that addresses the molecular mechanism by which Papi phosphorylation contributes to piRNA processing and supports the authors model.

Comments

- 1) The discovery of Papi-Ser547 and the identification of the responsible enzyme Par-1 is an important first step towards fully elucidating piRNA biogenesis. The obvious question is how does the

phosphorylation mechanistically enhance the RNA binding activity of Papi? A conformational change seems plausible, similar to the autoinhibitory effect described for other proteins. While structural studies are perhaps a long-shot and beyond the scope of this work, the authors should attempt to support the conformational change hypothesis by biochemical data, for example they could try crosslinking mass spec using the various Papi mutants. I would also like to see in vitro RNA binding assays with and without phosphorylation of Papi, by using the described mutants.

2) Another major question that this manuscript does not address is how do the piRNA precursors make it to mitochondria? Can the authors exclude that they are brought along by Siwi/Par-1 complexes? I noticed that in their model the authors have depicted Siwi/Par-1 as unloaded. Is there data to support this? If Siwi is indeed unloaded, then I wonder which factors are involved in transporting precursors to mitochondria and how the 5' ends that are then bound by Siwi are generated? In case the authors have good data to support either model this should be clearly stated in the text. To address these questions, I suggest that the authors repeat some of their experiments (Par-1 & Papi complex formation, RNA association) with a Siwi mutant that is loading incompetent to ensure they only assay properly loaded piRNA precursors.

3) In Fig4f-g the Papi1-480 mutant that the authors use shows multiple bands on the western blot, and thus does not seem to produce a stable protein. The authors should try to find a mutant that is not prone to degradation and then see if RNAs can be recovered. Perhaps one that is truncated directly after the Tudor domain would be suitable? If the same result is obtained using a stable protein this would make a stronger case for a conformational change that relaxes the inhibition of Papi's KH domain.

4) All Western blots probing for Papi show several bands which could be different phosphorylation states but also correspond to other isoforms. The authors should perform mass spec analysis of all three bands and confirm that these are indeed differentially phosphorylated forms of Papi. Related, based on the data in Fig1d the authors claim that piRNA populations are unaffected by S157A and S565A mutants, however this is not shown. The authors should probe this by sequencing the RNA recovered by CLIP as these mutants could also lead to promiscuous binding and piRNA processing.

5) The claim that dMLS is only weakly phosphorylated is not well supported by the presented data in Fig2a. The authors should quantify the different isoforms/phosphorylation states of Papi from replicate experiments. Are the changes statistically significant? Also in Fig2b it appears that the image was cropped very closely to the band showing dMLS Papi protein. Could the authors please provide uncropped images to better evaluate the data?

6) In Fig3h endogenous Siwi does indeed bind to the KH mutant of Papi but it does not appear to be as strong as the WT contrary to what is claimed by the authors in the text on page 7. Could the reduction in piRNAs and piRNA precursors (at least partially) be due to the reduced recovery of Siwi? The authors should quantify the observed changes in protein and RNA levels and perform appropriate statistical test.

6) What is known from the literature about the substrate specificity of Par-1? It would be interesting to

see a little more detailed characterisation of Par-1, specifically probing the localisation in BmN4 cells by immunofluorescence or identifying the targets by pulldown and mass spec. Was Par-1 recovered in mass spec experiments for Siwi? Are there existing data sets that could be mined?

7) Suggested changes to figures: It would be helpful to the reader to move up the schematic of Papi to panel 1a. Panel c of Fig1 should be moved to the supplement. Many figure panels do not show the ladder or size markers: 1d, 1e (WB), 2a, 2b, 2f, 2g, 2h (WB), 3a, 3b, 3c, 3d, 3e, 3f, 3h (WB), 4a (WB), 4b, and 4c. These should be added for clarity.

Reviewer #1

In this manuscript, the authors used MS to identify the phosphorylation sites of Papi. They found three potential phosphorylation sites and provided the MS/MS spectra of the corresponding phosphopeptides (Fig. 1c). However, I am not convinced by these spectra due to the following reasons:

1. All three peptides have missed cleavage sites, if these are real phosphopeptides, the authors should also see the phosphopeptide signals without miss cleavage sites, which should have better intensities and much easier to be identified.

We thank the reviewer for raising this point. The three phosphopeptides that we found in this study were (1) VKVES_{157(P)}PK, (2) SDRHPLSIS_{547(P)}NFDLSYPDPSR, and (3) NKQLNGS_{565(P)}DDFLHGER. We did not obtain the completely digested form of the first phosphopeptide (*i.e.*, VES_{157(P)}PK) because we set the minimum peptide length as 6. If the minimum length is ≤ 6 , the results will become too complex to interpret in this type of proteome analyses. The completely digested forms of the second and third phosphopeptides would have been HPLSIS_{547(P)}NFDLSYPDPSR and QLNGS_{565(P)}DDFLHGER. These peptides were not detected in this study, likely because trypsin missed some possible cleavage sites. We do not think we need to share this with this reviewer, but trypsin cleaves proteins/peptides at the C-terminal sides of lysine and arginine residues, but may miss lysine or arginine residues if they are close together. We repeated the experiments two more times and the data are shown in revised Supplementary Fig. S1b and S1c. Again, in these analyses, the completely digested forms of the phosphopeptides were not detected, but the data confirmed the phosphorylation of Papi at Ser157, Ser547, and Ser565. We moved the data in original Fig. 1c to revised Supplementary Fig. S1a in response to comments of Reviewer #2.

2. In these three MS/MS spectra, all the main/strongest peaks are not assigned, the assigned b and y fragments are relatively weak, which decrease the identification confidence.

We thank the reviewer for raising this point. One of the aims of this study was to identify the phosphorylation site in Papi that elicits the RNA-binding activity of Papi, even if the phosphorylation state was not dominant, and therefore we did not focus on the *b* and *y* fragments with the major/strongest peaks. In fact, our biochemical data firmly support the argument that “*Papi with phosphorylated Ser547 was rare in BmN4 cells.*” We had described this in the original text (page 13).

3. In MS/MS fragmentation, cleavage at N-terminal to proline usually generates the strongest b and y ions, but we could not see that phenomenon in the spectra of the peptides bearing S157 or S547.

We thank the reviewer for raising this point. We are also aware that proline often generates the strongest *b* and *y* ions, but to our knowledge this is not always the case. According to the data shown in the original Fig. 1c (revised Supplementary Fig. S1a) for the first phosphopeptide (VKVES_{157(P)}PK), the fragment ion PK was at *m/z* 244.22, which was not the strongest. For the second phosphopeptide (SDRHPLSIS_{547(P)}NFDLSYPDPSR), the fragment ion PDPSR was at *m/z* 571.24, which was the strongest within the peptide, and the fragment ion PSR was at *m/z* 359.27, which was relatively strong but not the strongest. The fragment ion PLSIS_{547(P)}NFDLSYPDPSR was at *m/z* 944.43, which was weakly detected.

According to the data shown in Supplementary Fig. S1b for the phosphopeptide (VKVES_{157(P)}PK), the fragment ion PK was at *m/z* 244.17, which was relatively strong but not the strongest. For the phosphopeptide (SDRHPLSIS_{547(P)}NFDLSYPDPSR), the fragment ion PDPSR was at *m/z* 571.26, which was the strongest within the peptide, and the fragment ion PSR was at *m/z* 359.23, which was relatively strong but not the strongest. The fragment ion PLSIS_{547(P)}NFDLSYPDPSR was not detected.

According to the data shown in Supplementary Fig. S1c for the phosphopeptide (SDRHPLSIS_{547(P)}NFDLSYPDPSR), the fragment ion PDPSR at *m/z* 571.27 and the fragment ion PSR at *m/z* 359.29 were relatively strong but not the strongest. The fragment ion PLSIS_{547(P)}NFDLSYPDPSR was at *m/z* 944.97, which was relatively weak. We believe that it is normal for the spectra to be complex like this, particularly when dealing with proteomic samples, because they contain a wide variety of ions.

Taken these into consideration, although later the authors provided evidence of phosphorylation on S547, it might not be the predominate phosphorylation site(s), as we could see two major Papi bands around 80 kDa, and also multiple bands in Papi1-480, suggesting other modification sites. Moreover, radioactive assay (Fig.2c) also indicated the predominate phosphorylated Papi is around 80 kDa. The authors may also want to analyze the two bands around 80 kDa but not 90 kDa, although the CILP showing the signal around 90 kDa.

We thank the reviewer for raising these points. The western blotting data shown in original Fig. 1b, for example, indicate that the Papi band of approximately 90 kDa is not abundant. However, our MS analysis showed that this band was phosphorylated at Ser157, Ser547, and Ser565 (original Fig. 1c and revised Supplementary Fig. S1a). We

confirmed this by repeating the analysis two more times (revised Supplementary Fig. S1b and S1c). Our CLIP data (original and revised Fig. 1d) clearly showed that Ser547, but not Ser157 or Ser565, phosphorylation of Papi was the key to elicit the RNA-binding activity of Papi. Although the Papi bands of approximately 80 kDa could be analyzed, it is possible to speculate from our experimental data that these bands are a mixture of Papi phosphorylated at Ser157, Ser547, and/or Ser565 in multiple combinations. It is also possible that these bands contain partially fragmented Papi at either the N- or C-terminus, or both. Considering all of this, we decided not to perform MS analysis to determine the phosphorylation state of the 80 kDa Papi bands because the results would be confusing and too complex to provide any solid conclusion.

Reviewer #2

Bombyx Papi facilitates Siwi-piRISC biogenesis via directly binding to both Siwi protein and Siwi-loaded piRNA precursors. Phosphorylation of Papi is required for its binding to piRNA precursors and crucial to Siwi-piRISC biogenesis. In this article, the authors studied how Papi is phosphorylated and how the phosphorylation regulates the RNA-binding activity of Papi for Siwi-piRISC biogenesis. First, the authors identified residue Ser547 as the critical phosphorylation site in Papi for Siwi-piRISC biogenesis. Next, they identified Par-1, located nearby the mitochondrial surface, as the kinase for phosphorylation of Papi. Their mechanistic studies showed that cytoplasmic Siwi recruits Par-1 to phosphorylate Papi, resulting in the alteration of the folding dynamics of Papi and in turn promoting Zuc-dependent Siwi-piRISC maturation. These findings are of high interest and I support their publication in Nature Communications. Most of the experiments are well done, and many of the data presented are convincing and support the conclusions. I only have a few comments to improve the manuscript, as follows.

We appreciate this very positive and encouraging comment.

1. In Fig 1d, the point mutation S157A in KH domain appears to cause a moderate reduction of RNA-binding capacity compared with wildtype Papi. In principle, removal of negative charged phosphorylation in KH domain should be in favor of RNA binding. How this mutation in the KH domain could affect the RNA-binding activity is not discussed. Also, it seems to me that the S565A mutant shows a similar reduction of phosphorylation. It's better to confirm these IB data using the Papi-pS547 monoclonal antibody.

We thank the reviewer for raising these points. We repeated the experiment three more times to confirm that the S157A and S565A mutations did not significantly reduce the

RNA-binding activity of Papi. The statistical data are shown in revised Fig. 1d. We also replaced the original data in Fig. 1d with new representative data (revised Fig. 1d). The possibility that the RNA-binding activity of Papi was affected by the S157A mutation and finding that this was not actually the case are described in the revised text (page 5). Western blotting using anti-Papi-pS547 monoclonal antibody confirmed that the S565A mutant was phosphorylated at Ser547, similar to WT Papi (revised Supplementary Fig. S2f). Notably, the S565A mutant migrated faster on the gel than WT Papi, which is why the topmost band of the S565A mutant was very weak in Fig. 1d (original and revised). The RNA-binding activity of this mutant was equivalent to that of WT Papi (see revised Fig. 1d). This finding is described in the revised text (page 6).

2. In Fig 2f, the phosphorylated Papi band is fuzzy. To validate the specificity of Par-1 to phosphorylation of Ser547, it's better to re-examine the effects of six kinases on Papi phosphorylation using the Papi-pS547 monoclonal antibody.

We thank the reviewer for raising this point. We immunoprecipitated Papi-F from all the samples described in Fig. 2f and probed them with anti-Papi-pS547 monoclonal antibody. The results are shown as revised Supplementary Fig. S2g and the text was modified accordingly (page 7). The data clearly indicate that Par-1, but not other kinase candidates, is responsible for Ser547 phosphorylation in Papi.

3. In Fig. 3e and 3f, the authors concluded that Siwi binds to Par-1 in the cytoplasmic fraction before piRNA precursors are loaded onto Siwi. I am wondering whether the mature Siwi-piRISC, which has been released from Papi and localized in cytoplasm, could still interact with Par-1.

We thank the reviewer for raising this point. We checked whether Par-1 could still bind to Siwi-piRISC and found that it did (see figure below). We also checked whether the Siwi 9RK mutant, which lacks Papi binding ability, bound to Par-1 and found that it did. This latter finding further supports our original idea that unloaded Siwi binds to Par-1 prior to Siwi-Papi association. We included this second set of data in revised Fig. 3f and modified the text accordingly (page 8). Although the result shown in the figure below is interesting, we did not include it in the revised manuscript, because we want to examine the physiological implications of Par-1 remaining with Siwi-piRISC in a future study.

4. *The labels on Western blotting panels should be improved.*

We thank the reviewer for raising this point. We have improved the labels of the western blotting panels as much as possible. We believe this has made them easier to read.

Reviewer #3

1) *The discovery of Papi-Ser547 and the identification of the responsible enzyme Par-1 is an important first step towards fully elucidating piRNA biogenesis. The obvious question is how does the phosphorylation mechanistically enhance the RNA binding activity of Papi? A conformational change seems plausible, similar to the autoinhibitory effect described for other proteins. While structural studies are perhaps a long-shot and beyond the scope of this work, the authors should attempt to support the conformational change hypothesis by biochemical data, for example they could try crosslinking mass spec using the various Papi mutants. I would also like to see in vitro RNA binding assays with and without phosphorylation of Papi, by using the described mutants.*

We thank the reviewer for raising these points. We tried hard to support the conformational change hypothesis by biochemical analysis, including crosslinking mass spectrometry, but failed for two main possible reasons. First, the dynamics and/or turnover of Papi is likely to be too fast for us to be able to obtain enough protein for an assay. Second, we know that the high stability and high expression level of Siwi in BmN4 cells make it difficult to knockdown Siwi to a satisfactory degree by RNAi, and therefore residual Siwi would interfere as noise for mutant analysis. We attempted to partially digest the Papi mutants (S547A and S547D/E) *in vitro*. We previously performed a similar experiment and successfully showed structural changes of Piwi (one of the *Drosophila* orthologs of silkworm Siwi) before and after piRNA loading (Yashiro et al. *Cell Rep*.

23:3647–3657, 2018). However, the experiment with Papi was not successful. This was most likely because of the multiple intrinsic disordered regions (IDRs) found in Papi (original Supplementary Fig. S4h). We also tried to conduct a structural analysis of Papi mutants (S547A and S547D/E) with the help of a long-time collaborator, Prof. Hiroshi Nishimasu at The University of Tokyo. However, Prof. Nishimasu informed us that he had no luck so far; cryo-electron microscopy failed to determine the Papi structure. He thought that this was likely because of the IDRs in Papi. Considering all these findings, we decided to modify our original model (original Supplementary Fig. S4b–f) to tone down our discussion (revised Supplementary Fig. S4b and page 11 in the revised text). However, we assert that this modification does not affect the importance of our findings. We also note that the revised model follows the design of our previous model (Fig. 1i in Nishida et al. *Nature* 2018; ref#14 in the revised manuscript).

As we described in the original Introduction (page 2), our findings show that the RNA-binding activity of Papi was regulated through a very sophisticated mechanism that involved unloaded Siwi and its loading with piRNA precursors. The finding that the Papi1–222 mutant was able to bind RNAs *in vivo*, whereas the Papi1–480 mutant was unable to, also illustrates the complexity of the mechanism. While both mutants lack Ser547 phosphorylation, the Papi1–222 mutant, but not Papi1–480 mutant, binds to Siwi. To reproduce this *in vitro*, we need to prepare the appropriate amount of unloaded Siwi. However, this is very difficult to do because unloaded Siwi is very unstable. This problem is widely recognized in piRNA field.

2) *Another major question that this manuscript does not address is how do the piRNA precursors make it to mitochondria? Can the authors exclude that they are brought along by Siwi/Par-1 complexes? I noticed that in their model the authors have depicted Siwi/Par-1 as unloaded. Is there data to support this? If Siwi is indeed unloaded, then I wonder which factors are involved in transporting precursors to mitochondria and how the 5' ends that are then bound by Siwi are generated? In case the authors have good data to support either model this should be clearly stated in the text. To address these questions, I suggest that the authors repeat some of their experiments (Par-1 & Papi complex formation, RNA association) with a Siwi mutant that is loading incompetent to ensure they only assay properly loaded piRNA precursors.*

We thank the reviewer for raising these points. The interaction between unloaded Siwi and Papi requires symmetrical dimethylarginine modification of Siwi. The Siwi 9RK mutant that lacks the symmetrical dimethylarginine modification has been shown not to

bind Papi, and the mutant was free from piRNA precursor (Nishida et al. *Nature* 2018; ref#14 in the revised manuscript). On the basis of these observations, we argue that Siwi is loaded with piRNA precursor only after it binds to Papi. We described this in the revised text (page 33). We have recently shown that the Siwi 9RK mutant can bind Par-1 (revised Fig. 3f), which further supports our original idea that Siwi and Par-1 bind in the cytoplasm before Siwi binds to Papi. The text was modified accordingly (Page 8).

Previously, we have shown that Siwi-piRISC was hardly assembled in the absence of Spindle-E (Spn-E) and that Spn-E binds to unloaded Siwi (Nishida et al. *Cell Rep.* 2015; ref#6 in the revised manuscript). On the basis of these observations, we assumed that Spn-E is the factor that provides piRNA precursor to unloaded Siwi upon its binding to Papi. We are currently analyzing this hypothesis experimentally, and hope to publish the results in a separate paper. We currently have no idea how the 5' ends of piRNA precursors are generated prior to their binding to Siwi. We are currently addressing this important question and willing to report the results in the future.

3) In Fig4f-g the Papi1-480 mutant that the authors use shows multiple bands on the western blot, and thus does not seem to produce a stable protein. The authors should try to find a mutant that is not prone to degradation and then see if RNAs can be recovered. Perhaps one that is truncated directly after the Tudor domain would be suitable? If the same result is obtained using a stable protein this would make a stronger case for a conformational change that relaxes the inhibition of Papi's KH domain.

We thank the reviewer for raising these points. Despite our best efforts, we have not been able to produce more stable mutants. The mutant that was truncated directly after the Tudor domain was even more unstable. Because Ser157 is present in the Papi1–480 mutant sequence, we treated this mutant with phosphatase. This resulted in the disappearance of the upper two bands on the western blot gel of this mutant as well as that of WT Papi. Therefore, it is reasonable to assume that these bands correspond to phosphorylated Papi1–480 at Ser157. The lower of the two bands, which had a slightly weaker signal than that of the upper band, may be a degraded form of the upper band. In addition, an *in vitro* translation system was used to translate the Papi1–480 mutant. This system does not allow phosphorylation to occur. The product appeared as a single band, which comigrated with the Papi1–480 signal at approximately 60 kDa on the gel. This corroborates that the upper two bands of the mutant are phosphorylated Papi1–480. These new data are shown as revised Supplementary Fig. S4c and S4d and the text

was modified accordingly (page 10). We also noted in the revised legend of Fig. 4f that the top two bands are phosphorylated forms of the Papi mutant.

4) All Western blots probing for Papi show several bands which could be different phosphorylation states but also correspond to other isoforms. The authors should perform mass spec analysis of all three bands and confirm that these are indeed differentially phosphorylated forms of Papi. Related, based on the data in Fig1d the authors claim that piRNA populations are unaffected by S157A and S565A mutants, however this is not shown. The authors should probe this by sequencing the RNA recovered by CLIP as these mutants could also lead to promiscuous binding and piRNA processing.

We thank the reviewer for raising these points. In our previous study, we showed that Papi appeared as a single band upon dephosphorylation (Fig. 1h in Nishida et al. *Nature* 2018; ref#14 in the revised manuscript). That band corresponds to the lowest of the multiple bands of Papi (original Fig. 1a). This strongly supports the idea that the upper bands of Papi are phosphorylated Papi, and not isoforms. In this study, we found that Papi phosphorylation at Ser547 by Par-1 is necessary for Papi to promote Siwi-piRISC biogenesis in BmN4 germ cells. The Papi S157A and S565A mutants were able to bind RNAs, similar to WT Papi (revised Fig. 1d). Western blotting using anti-Papi-pS547 monoclonal antibody confirmed that the S157A and S565A mutants were phosphorylated at Ser547, similar to WT (revised Supplementary Fig. S2e). The other Papi bands could also be analyzed. However, from our experimental data, it is possible to speculate that these bands are a mixture of Papi phosphorylated at Ser157, Ser547, and/or Ser565 in multiple combinations. It is also possible that these bands contain partially fragmented Papi at either the N- or C-terminus, or both. Considering these, we decided not to perform MS/MS analysis to determine the phosphorylation state of the 80 kDa Papi bands because the results would be too complex to provide any solid conclusion.

The S157A and S565A mutations may lead to promiscuous RNA binding. We are interested in investigating the biological meanings of Ser157 and Ser565 modifications and identifying other kinases in future studies.

5) The claim that dMLS is only weakly phosphorylated is not well supported by the presented data in Fig2a. The authors should quantify the different isoforms/phosphorylation states of Papi from replicate experiments. Are the changes statistically significant? Also in Fig2b it appears that the image was cropped very

closely to the band showing dMLS Papi protein. Could the authors please provide uncropped images to better evaluate the data?

We thank the reviewer for raising these points. The original western blotting data shown in Fig. 3a confirmed that the Papi Δ MLS mutant was not phosphorylated at Ser547. We also quantified the signals of the phosphorylated Δ MLS mutant, and found that they were weak (revised Supplemental Fig. S2b). The revised Fig. 2b has an uncropped image, which was the original data obtained in the experiment.

6) In Fig3h endogenous Siwi does indeed bind to the KH mutant of Papi but it does not appear to be as strong as the WT contrary to what is claimed by the authors in the text on page 7. Could the reduction in piRNAs and piRNA precursors (at least partially) be due to the reduced recovery of Siwi? The authors should quantify the observed changes in protein and RNA levels and perform appropriate statistical test.

We thank the reviewer for raising this point. We repeated the experiments three times and provided the statistics in revised Supplementary Fig. S3. We would like to note that this validation had no influence on our original claim. The representative gel image is now shown in revised Fig 3i.

7) What is known from the literature about the substrate specificity of Par-1? It would be interesting to see a little more detailed characterisation of Par-1, specifically probing the localisation in BmN4 cells by immunofluorescence or identifying the targets by pulldown and mass spec. Was Par-1 recovered in mass spec experiments for Siwi? Are there existing data sets that could be mined?

We thank the reviewer for raising this point. As noted in the original Discussion, Par-1 is known to phosphorylate a number of proteins. A previous review paper of Par-1 in mammals reported a consensus sequence (ref#33 in the revised manuscript). However, we did not find a consensus-like sequence in silkworm Papi. We have noted this in the revised Discussion (page 13).

The data in original Fig. 3b show that Par-1 is present in the cytoplasm of BmN4 cells. The mitochondrial fraction also contained Par-1, but the signal was weak. Par-1 in *Drosophila* cells seemed to be cytoplasmic (Bayraktar et al. *Journal of Cell Science* 19: 711–721, 2006). Immunofluorescence may help to determine the subcellular localization of Flag-Par-1 that was exogenously expressed in BmN4 cells, but this issue seems less important to examine in the current study. We are also interested in investigating other substrates of Par-1 in BmN4 cells, and hope to be able to conduct pull-down and mass spectrometry in the future.

8) *Suggested changes to figures: It would be helpful to the reader to move up the schematic of Papi to panel 1a. Panel c of Fig1 should be moved to the supplement. Many figure panels do not show the ladder or size markers: 1d, 1e (WB), 2a, 2b, 2f, 2g, 2h (WB), 3a, 3b, 3c, 3d, 3e, 3f, 3h (WB), 4a (WB), 4b, and 4c. These should be added for clarity.*

We thank the reviewer for these suggestions. We kept the schematic diagram of Papi as revised Fig. 1c because it shows the phosphorylation sites identified in the MS analysis. The MS data in original Fig. 1c were moved to Supplementary Fig. S1a, but the schematic diagram was kept as Fig. 1c in the revised manuscript. We added the ladder and size markers where possible.

REVIEWERS' COMMENTS

Reviewer #2 (Remarks to the Author):

All my concerns have been fully and detailedly addressed in the revised manuscript. And I recommend its publication in Nature Communications.

Reviewer #3 (Remarks to the Author):

This revised manuscript has been improved and the authors have addressed most of the reviewers' concerns. There are no further technical concerns about the content.